behaviour/biomechanics

feather loss, wing kinematics, whole-body kinematics, flapping performance, bird, keel bone damage

**Author for correspondence:**
Bret W. Tobalske
e-mail: bret.tobalske@mso.umt.edu

# Domestic egg-laying hens, *Gallus gallus domesticus*, do not modulate flapping flight performance in response to wing condition

Brianna M. León[1], Bret W. Tobalske[2], Neila Ben Sassi[1], Renée Garant[1], Donald R. Powers[3] and Alexandra Harlander-Matauschek[1]

[1]Department of Animal Biosciences, University of Guelph, 50 Stone Road E, Guelph, Ontario, Canada N1G 2W1
[2]Division of Biological Sciences, University of Montana, 32 Campus Drive, Missoula, MT 59812, USA
[3]Department of Biology, George Fox University, 414 N Meridian Street, Newberg, OR 97132, USA

DRP, 0000-0003-1126-7141; AH-M, 0000-0001-6869-5281

Wild birds modulate wing and whole-body kinematics to adjust their flight patterns and trajectories when wing loading increases flight power requirements. Domestic chickens (*Gallus gallus domesticus*) in backyards and farms exhibit feather loss, naturally high wing loading, and limited flight capabilities. Yet, housing chickens in aviaries requires birds to navigate three-dimensional spaces to access resources. To understand the impact of feather loss on laying hens' flight capabilities, we symmetrically clipped the primary and secondary feathers before measuring wing and whole-body kinematics during descent from a 1.5 m platform. We expected birds to compensate for increased wing loading by increasing wingbeat frequency, amplitude and angular velocity. Otherwise, we expected to observe an increase in descent velocity and angle and an increase in vertical acceleration. Feather clipping had a significant effect on descent velocity, descent angle and horizontal acceleration. Half-clipped hens had lower descent velocity and angle than full-clipped hens, and unclipped hens had the highest horizontal acceleration. All hens landed with a velocity two to three times greater than in bird species that are adept fliers. Our results suggest that intact laying hens operate at the maximal power output supported by their anatomy and are at the limit of their ability to control flight trajectory.

# 1. Introduction

Feathers are an essential and defining aspect of a bird's biology as they provide insulation, waterproofing and grant most birds with flight capabilities [1]. Any loss or damage to flight feathers of the wing (remiges) has the potential to adversely impact flight performance [2]. However, feathers are dead structures, and they naturally and quickly become damaged and degrade, making the need to replace damaged feathers essential to survival [1,3]. Thus, most bird species undergo at least a yearly, energy-demanding moult period, where old, worn feathers are slowly replaced with new feathers [1,4]. Wing feathers moult in two main fashions. A rapid synchronous moult, seen in waterfowl, renders the birds flightless for several weeks [5]. More commonly, old wing feathers are gradually and symmetrically pushed out by newly forming feathers, ensuring birds retain some flight capabilities to forage and escape predators [1,6].

Impaired flight ability due to feather loss or damage may particularly affect domestic backyard and commercially farmed chickens (*Gallus gallus domesticus*) kept for egg-laying purposes. Along with the natural moult, feather loss and damage due to bird-to-bird pecking occurs in 15–95% of these birds [7]. In these environments, birds must inevitably navigate around various structures to access food and water or escape in response to a threat [8]. Theoretically, impaired flight control can lead to keel bone fractures. The keel is the protruding feature of the sternum, the location where the main flight muscles originate [8,9]. Despite being detected in up to 97% of laying hens [10], the causes for keel bone injuries are still unknown, and current routes of investigation include genetic predisposition, housing adaptations and development, nutrition, high-energy dynamic events such as collisions, and low-energy non-collision static events [8]. The variety of investigative routes emphasizes the severity of finding a solution to this significant and painful health issue for birds worldwide [11–13]. Stratmann *et al.* [14] found that breeds of laying hens with lower wing loading demonstrated fewer incidences of keel bone fractures and deviations compared with breeds with higher wing loadings. Thus, an increase in wing loading through feather loss may lead to keel bone injury as the birds may have greater difficulty manoeuvring, which may increase their rate of high-energy dynamic events [15]. Although high-energy collisions may only be responsible for a small subset of keel bone fractures [16], this potential source of injury still merits investigation.

Slow-speed flight (i.e. after take-off and before landing) requires great power output [17], and the need to control manoeuvring around obstacles increases this power cost [18,19]. Meanwhile, the loss of the wings' surface area increases wing loading (body weight per unit wing area), which should increase induced power requirements (the power required to produce lift for weight support and thrust). Induced power dominates total power required during slow flight [20,21]. Thus, it is expected that a reduction in wing area will diminish the control of manoeuvring performance in slow flight unless a bird has reserves capable of increasing power output from its flight muscles and successfully transferring this power to the air.

This increase in power output can be mediated by increasing a combination of wingbeat kinematic parameters such as wingbeat amplitude and frequency to varying extents [22–25]. Understanding the changes to flapping patterns and wing kinematics that result from increased wing loading would elucidate how aerial birds' manoeuvrability is affected by wing feather loss. Indeed, studies show that increased power output is required to maintain performance in response to increased wing loading [26,27].

The ability to manoeuvre during slow flight should decrease as a function of increasing body size among bird species, in part because wing loading scales positively with body mass and, more significantly, the power available from the flight muscles scales negatively, approximately proportional to wingbeat frequency [28–31]. Several bird species with low wing loading, such as the goldcrest and coal tit, exhibit greater agility and manoeuvrability at lower flight speeds [32]. As size increases among bird species, the difference between power available from the muscles and the power required for flight will decrease [28,33]. Hummingbirds, being the smallest birds, are thus the most manoeuvrable birds [34,35]. In sum, higher loading and lower marginal power available for manoeuvring should limit larger birds' capacity to control their trajectory during slow-speed flight.

Previous studies assessing the kinematic effects of wing loading changes have primarily focused on species that use flight as a primary means of locomotion. However, ground-dwelling birds such as pheasants, chickens and other Phasianidae perform flight mainly to escape predators and to roost. They perform a quick, explosive take-off that requires high power, anaerobic output, then rapidly return to the ground to run or hide [29,36]. These species exhibit unusually high wingbeat frequencies for their body size, consistent with high wing loading and the high power required for take-off [37].

Therefore, feather damage may be particularly troublesome for Phasianidae birds due to limited power availability for controlling flight trajectory.

Our research question aims to determine whether symmetric wing feather loss impacts flapping-flight performance in domestic, egg-laying chickens, which, in turn, may implicate feather loss as a risk factor for keel bone injuries. We used adult hens with three levels of feather clipping: unclipped, half-clipped (only primary feathers clipped) and full-clipped (both primary and secondary feathers clipped). We measured wing and body kinematics during descent from a platform to the ground to assess flapping-flight performance. We used wingbeat frequency, amplitude and angular velocity as power output indicators [17,38,39]; vertical and horizontal acceleration, descent angle and descent velocity as flight performance metrics; and bilateral kinematic asymmetries as indices of control [40]. We predicted that progressive increases in wing loading due to feather clipping would cause birds to compensate with their wing motions to accomplish similar body trajectories. This would indicate a greater power requirement for flapping-flight performance in birds with greater wing loading, and if they are not able to accomplish similar body trajectories, a reduced ability to offset gravitational forces that could result in high-energy events.

# 2. Material and methods

## 2.1. Study animals and housing

A total of 18 adult domestic, white-feathered laying hens (Lohmann LSL lite) 34 weeks of age were used in this study. They were housed in six aviary-style pens, with a total of 10 hens per pen. Three hens from each pen participated in this study. The pens (183 $L \times 244$ $W \times 290$ $H$ cm) were covered in 5 cm of wood shavings with a feeder in the middle of the pen and a nest-box against the rear wall. Two platforms (122 $L \times 31$ $W$, 70 cm above the ground) were placed on either side of the pen, with a second nest-box on one platform and a second feeder on the other. The pens were also equipped with a high perch (5 $\varnothing$ at 150 cm above the ground) towards the pen's rear, spanning the pen's width. Automatic drinkers were placed beneath one of the platforms. The room that housed the hens averaged 21°C and had a 14 : 10 h light : dark light cycle with a 30 min dawn and dusk period. All the hens were part of a larger ongoing experiment assessing keel bone injury and the growth/shrinkage of pectoralis and leg muscle in response to wing clipping (R.G. *et al*. 2020, unpublished observations).

## 2.2. Jump tower apparatus

The testing apparatus to evaluate flight movements was erected in a test arena (366 $L \times 244$ $W \times 290$ $H$ cm) in a separate room. A jump tower constructed of wood and steel was placed on one end of the arena. The jump tower had slots running vertically so that a start box and platform assembly could be inserted, and the height could be adjusted from 10 to 170 cm in 20 cm increments. The start box and platform assembly (figure 1) were placed at the height of 150 cm during all training and testing protocols, the same height as the high perch in the home pens. The start box (30 $W \times 30$ $L \times 45$ $H$ cm, interior dimensions) was constructed of lightweight plywood and metal mesh for the ceiling. The front of the start box had a corrugated plastic sliding door that ensured the hen travelled in only one direction. The hen could exit the start box by stepping onto a lightweight plywood platform (20 $L \times 30$ $W$ cm) from which she could jump down onto the test arena. The hens were allowed time to acclimatize to being placed on and jumping down from the tower apparatus. Most hens descended immediately after being placed in the start box.

The floors of the testing arena were covered with foam interlocking mats (Interlocking play foam floor tiles, Canadian Tire®, Guelph, Ontario, Canada). A container of sweet corn kernels and dead mealworms was placed on the pen floor across from the jump tower to entice the birds. Two high-speed video cameras (GoPro Hero6® and GoPro Hero7®, GoPro, San Mateo, CA, USA) were used to record the birds. They recorded 240 frames s$^{-1}$ with the pixel ratio set to 'Narrow View' 1920 × 1080 pixels and the ProTune option set to high shutter speed. One camera was placed on the wall directly across from the jump tower 2.5 m from the ground, above the corn and mealworm container, to capture a frontal view. The second camera was placed 2.5 m from the ground at the midpoint of the wall perpendicular to the jump tower to capture a lateral view. This frontal and lateral view recorded the bird's jump simultaneously. A metre stick was placed on the wall opposite the lateral camera in the recording field of view.

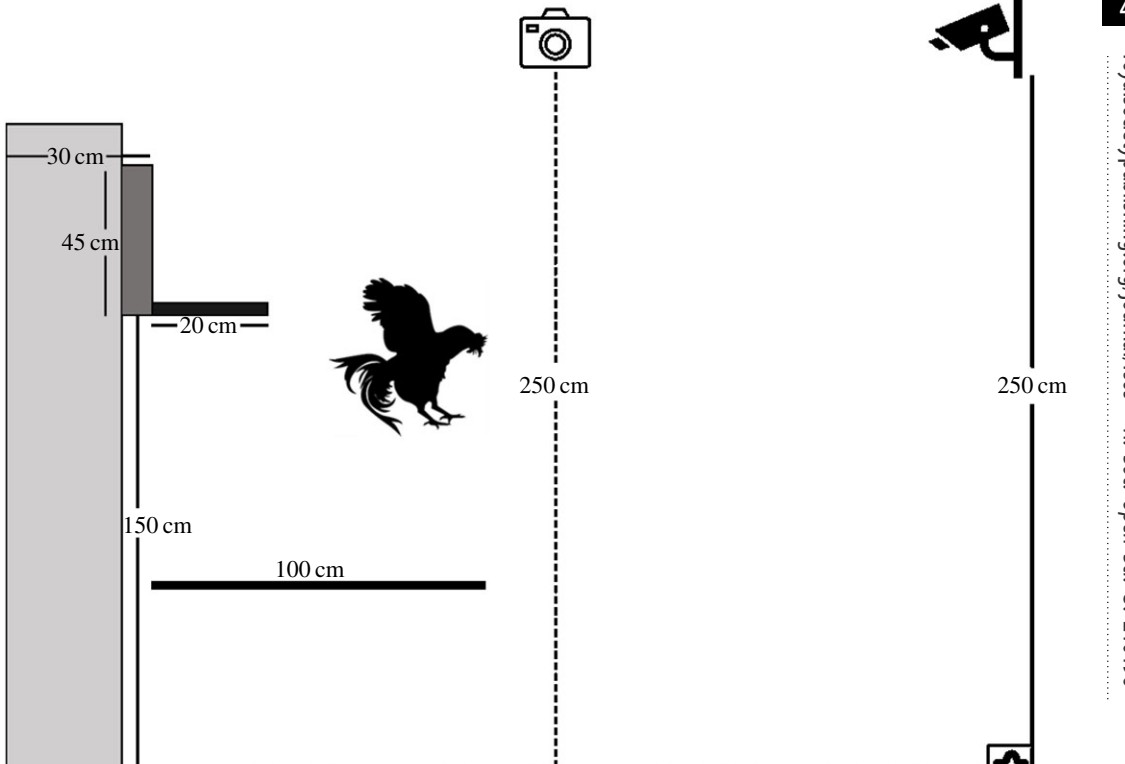

**Figure 1.** Testing jump tower apparatus with start box and platform. The frontal camera is indicated by a solid figure and the lateral camera is indicated by the outlined figure.

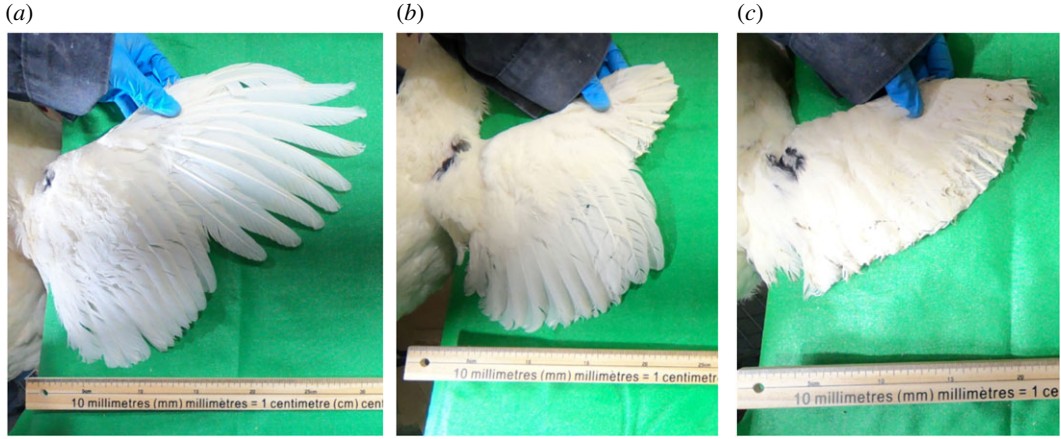

**Figure 2.** Wing clipping treatments of the hens against a solid background and a metre stick to measure fixed distance. (*a*) The unclipped hens retained all their primary and secondary feathers; (*b*) half-clipped hens had their 10 primary feathers clipped on each wing; (*c*) full-clipped hens had their primary and secondary feathers clipped on each wing.

## 2.3. Wing treatment and morphological measurements

The 18 birds used in the study were randomly distributed into one of three treatment groups (figure 2):

1. Unclipped: no wing feathers clipped.
2. Half-clipped: symmetrical clipping of the 10 primary feathers.
3. Full-clipped: symmetrical clipping of all primary and secondary feathers.

Each of the six pens housed one hen from each treatment group to balance the pens throughout the housing room. Each bird had its left and right wings photographed with a GoPro Hero6® (GoPro, San

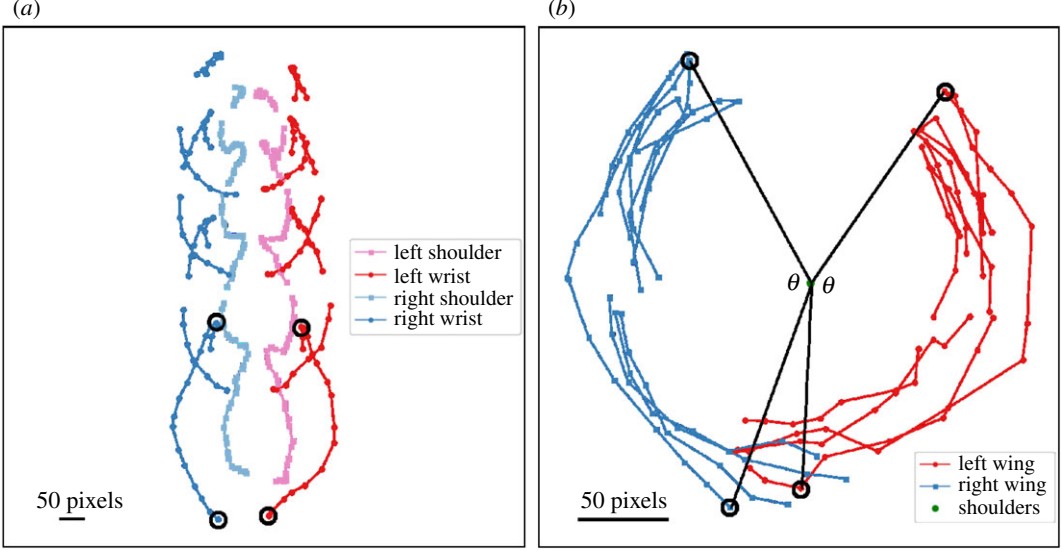

**Figure 3.** Two-dimensional wing kinematics from a hen (*Gallus domesticus*) engaged in descending flight. (*a*) Frontal view of shoulders and wrists in global reference frame. (*b*) Frontal view of wrist points in shoulder-centred reference frame.

Mateo, CA, USA). Wings were positioned with the ventral side facing down, and each wing spread out against a solid background with a metre stick in view (figure 2). Birds were weighed (kg) with a digital scale (Ohaus™ Ranger™, Fisher Scientific).

## 2.4. Testing procedure

We moved hens from their home pen and placed them in the test arena. The shoulder feathers were marked with a 3 cm dot using a non-toxic black marker to allow easy spotting in the video recordings. The bird was then placed into the start box and platform assembly. The bird jumped down from the platform (figure 1) three times while flapping her wings to ensure that a straight flight path was recorded. After each jump, the bird was rewarded with sweet corn and dead mealworms. The bird returned to the home pen once all three jumps were performed.

## 2.5. Wing and whole-body kinematics calculations and statistical analysis

For the straightest jump recorded by each hen, we tracked the left and right shoulders and wrists from the frontal view and the right eye from the lateral view using the Matlab script DLTdv8a [41]. All birds had a downward descent and did not jump upwards before descending. We tracked the shoulder and wrist points in the 10–20 frames surrounding the highest point of the upstroke and the lowest point of the downstroke in their aerial descent from the frontal view. The first downstroke was defined as the first complete stroke of the wings towards the floor after the hen's feet had left the jump platform. The final downstroke was defined as the last complete stroke of the wings towards the floor before the hen landed on the ground. From the lateral view, we tracked the hen's right eye at the three single frames corresponding with the highest points of their first and second upstroke and the lowest point of their last downstroke.

All calculations and wingbeat visualizations were completed using Matlab (figure 3). The frontal view allowed us to calculate the wingbeat amplitude ($\Theta$), frequency (Hz) and angular velocity of the wrist (rad s$^{-1}$). Amplitude was measured as the angle in degrees covered by the wings from the top of the upstroke to the bottom of the downstroke, using the line between the shoulder and the wrist (figure 3). The frontal-view camera was approximately parallel to the long-axis of each hen's body and orthogonal to the flight trajectory. However, variation in angles of the stroke planes and descent contributed to unknown error in our measurement of $\Theta$. Our lateral-view measures of descent angle ($\gamma$) suggest this error was less than or equal to 4%. Wingbeat frequency was calculated as the frames per second of the video recording divided by the number of frames between the beginning of one downstroke to the beginning of the next downstroke. Angular velocity was calculated by converting the amplitude to radians and dividing by the downstroke duration in seconds. Wingbeat amplitude,

frequency and angular velocity were calculated as the mean of the left- and right-wing measures within each bird for all wingbeats performed in the jump. We also calculated the asymmetry of wingbeat amplitude, frequency and angular velocity, defined as the absolute value of the difference between the left- and right-wing measures within each bird for all wingbeats performed in the jump.

The lateral view allowed us to calculate whole-body kinematics. The metre stick in the field of view was first used to create a pixels-to-metric conversion. Descent velocity (m s$^{-1}$) was calculated by taking the first and third positions of the eye and a standardized point in the field of view to compute the distance travelled, which in turn, was divided by the total frames between the two points. Descent angle ($\gamma$) was calculated by using the dot product between a horizontal position vector and a position vector obtained by subtracting the first from the third position of the eye. Vertical acceleration (m s$^{-2}$) was calculated by dividing the vertical velocity difference between the second to third eye positions and the first to second positions by the difference in average time between the second to third positions and the first to second positions. Horizontal acceleration (m s$^{-2}$) was calculated by dividing the horizontal velocity (m s$^{-1}$) difference between the second to third eye positions and the first to second positions by the difference in average time between the second to third positions and the first to second positions.

To calculate wing loading, each wing's outline was first traced using ImageJ software to determine the wing area (cm$^2$) for each wing. Wing loading (N m$^{-2}$) was then calculated as body mass (kg) times gravitational acceleration (g; 9.81 m s$^{-2}$) divided by the sum of the left- and right-wing areas (m$^2$) [42]. For all wing kinematics measured in this study, we computed a mean within each individual hen. These within-individual means were then used to calculate the mean ± s.e. for each treatment group. This was done to balance the fact that most hens performed three complete wingbeats in their jumps, whereas some birds performed two or four complete wingbeats in their jumps.

Statistical tests were conducted using the generalized linear mixed model procedure (PROC GLIMMIX) in SAS Studio University Edition v. 9.04 (2012; SAS Institute Inc., Cary, NC, USA). Studentized residuals plots were used to test for normality of the data and determine the best-suited distribution.

Wingbeat frequency (Hz), asymmetry of amplitude ($\Theta$), asymmetry of frequency (Hz), asymmetry of angular velocity (rad s$^{-1}$), descent velocity and vertical acceleration were transformed into a lognormal distribution. All other outcome variables were kept as a Gaussian distribution, except for the number of wingbeats, which was a Poisson distribution. Hen weight ($N$) was added as a covariate for all outcome measures. Pen number was added as a random effect to test for any block effects, except in the case of count data number of wingbeats performed, to test for a pen location effect. Estimates for all outcome measures were calculated using a lsmeans statement and a Tukey–Kramer adjustment for multiple comparisons.

The lateral footage used for two hens (one unclipped and one half-clipped) was lost due to a camera battery complication. Thus, descent velocity (m s$^{-1}$), descent angle ($\gamma$) and horizontal and vertical acceleration (m s$^{-2}$) have an $N = 5$ for the unclipped and half-clipped treatment groups.

## 3. Results

The means ± s.e. for all wing and whole-body kinematic measures are listed in tables 1 and 2, respectively, for unclipped, half-clipped and full-clipped birds. As expected, wing clipping significantly increased wing loading between unclipped versus half-clipped hens ($p = 0.0285$) and half-clipped versus full-clipped hens ($p < 0.0001$). Clipping the primary feathers resulted in a 32.54% loss in the wing area. Clipping both the primary and secondary feathers resulted in a 55.37% loss in the wing area. Interestingly, we report that clipping status did not significantly affect the number of wingbeats performed (table 3). Overall, the hens performed three wingbeats as they descended from the 1.5 m tall platform.

Wing clipping status did not significantly affect the wingbeat amplitude ($\theta$), frequency or angular velocity (table 3). Although there was an almost 10° difference in $\theta$ between the unclipped and fully clipped birds, this difference was not significant due to the dataset's high standard error (table 1). Asymmetry values were used to detect any significant differences between right and left wings' performance relative to clipping status. The asymmetries of $\theta$, frequency and angular velocity were not significantly affected by the wing clipping treatments (table 3). Body weight was a significant covariate in determining the asymmetry of angular velocity (table 3); however, it did not contribute to significant differences among hens of different clipping statuses (table 4). Although the mean asymmetry of amplitude is almost double in the unclipped and full-clipped hens compared to the

**Table 1.** Mean ± s.e. of wing kinematic measurements based on clipping status (unclipped, half-clipped—primary feathers clipped, full clipped—primary and secondary feathers clipped). Frequency, amplitude and angular velocity values are the average of left- and right-wing measures for each bird within each group. Asymmetry measures represent the absolute value of the difference between left- and right-wing measures.

| wing kinematic measures | unclipped (N = 6) | half-clipped (N = 6) | full-clipped (N = 6) |
|---|---|---|---|
| number of wingbeats | 3.00 ± 0.71 | 3.17 ± 0.73 | 3.17 ± 0.73 |
| amplitude ($\Theta$) | 140.72 ± 6.25 | 138.87 ± 5.76 | 131.60 ± 5.56 |
| frequency (Hz) | 9.88 ± 0.382 | 10.31 ± 0.35 | 9.61 ± 0.31 |
| angular velocity (rad s$^{-1}$) | 46.75 ± 2.06 | 49.49 ± 1.86 | 47.50 ± 1.78 |
| asymmetry of amplitude ($\Theta$) | 17.45 ± 7.30 | 8.40 ± 3.13 | 15.29 ± 5.42 |
| asymmetry of frequency (Hz) | 0.00 ± 0.01 | 0.73 ± 1.51 | 0.01 ± 0.02 |
| asymmetry of angular velocity (rad s$^{-1}$) | 12.70 ± 5.69 | 3.30 ± 1.30 | 4.99 ± 1.86 |

**Table 2.** Mean ± s.e. of whole-body kinematic measurements based on clipping status (unclipped, half-clipped—primary feathers clipped, full clipped—primary and secondary feathers clipped).

| whole-body kinematic measures | unclipped (N = 5) | half-clipped (N = 5) | full-clipped (N = 6) |
|---|---|---|---|
| descent velocity (m s$^{-1}$) | 4.15 ± 0.38 | 3.24 ± 0.35 | 4.44 ± 0.31 |
| descent angle ($\gamma$) | 54.75 ± 4.94 | 43.27 ± 4.49 | 59.05 ± 3.99 |
| vertical acceleration (m s$^{-2}$) | 2.64 ± 0.95 | 4.63 ± 0.86 | 2.10 ± 0.76 |
| horizontal acceleration (m s$^{-2}$) | −0.78 ± 0.36 | −3.35 ± 0.32 | −3.38 ± 0.29 |

**Table 3.** Effect of clipping status as a fixed effect and body weight as a covariate on wing kinematics.

| wing kinematic measures | clipping status | | body weight | |
|---|---|---|---|---|
| | F-value | p-value | F-value | p-value |
| number of wingbeats | 0.01 | 0.9855 | 0.20 | 0.6663 |
| amplitude ($\Theta$) | 1.44 | 0.2877 | 0.19 | 0.6788 |
| frequency (Hz) | 1.24 | 0.3340 | 0.26 | 0.6194 |
| angular velocity (rad s$^{-1}$) | 0.74 | 0.5059 | 0.04 | 0.8390 |
| asymmetry of amplitude ($\Theta$) | 1.22 | 0.3396 | 3.28 | 0.1036 |
| asymmetry of frequency (Hz) | 1.48 | 0.2791 | 2.81 | 0.1279 |
| asymmetry of angular velocity (rad s$^{-1}$) | 1.83 | 0.2151 | 7.20 | 0.0250* |

*Significance at $p < 0.05$.

half-clipped hens, these differences were not statistically significant as the standard errors were relatively high (table 1). A similar pattern of means and standard errors is observed for the asymmetry of angular velocity (table 1).

Symmetric wing clipping had a significant effect on descent velocity (table 5), which averaged 3.94 m s$^{-1}$ across all hens. Descent angle ($\gamma$) was also significantly impacted by clipping status (table 5), with the half-clipped hens descending at a lower $\gamma$ than their full-clipped counterparts (table 6). Surprisingly, the unclipped and full-clipped hens had similar descent velocities and $\gamma$, where the half-clipped hens exhibited lower descent velocities and $\gamma$ than the full-clipped hens (table 6). Comparing the first and second halves of the descent, the horizontal component of descent velocity decreased by 0.42–1.87 m s$^{-1}$ for all birds. However, the vertical component of descent velocity remained constant for the unclipped and full-clipped hens and increased by 2.13 m s$^{-1}$ in the

**Table 4.** Comparison of wing kinematics between unclipped, half-clipped and full-clipped groups.

| wing kinematic measures | t-value | p-value |
|---|---|---|
| number of wingbeats | | |
| unclipped versus half-clipped | 0.17 | 0.9844 |
| unclipped versus full-clipped | 0.14 | 0.9898 |
| half-clipped versus full-clipped | −0.05 | 0.9984 |
| amplitude ($\Theta$) | | |
| unclipped versus half-clipped | 0.26 | 0.9643 |
| unclipped versus full-clipped | 1.35 | 0.4039 |
| half-clipped versus full-clipped | 1.38 | 0.3915 |
| frequency (Hz) | | |
| unclipped versus half-clipped | −0.72 | 0.7562 |
| unclipped versus full-clipped | 0.51 | 0.8670 |
| half-clipped versus full-clipped | 1.58 | 0.3041 |
| angular velocity (rad s$^{-1}$) | | |
| unclipped versus half-clipped | −1.04 | 0.5714 |
| unclipped versus full-clipped | −0.30 | 0.9503 |
| half-clipped versus full-clipped | 1.03 | 0.5799 |
| asymmetry of amplitude ($\Theta$) | | |
| unclipped versus half-clipped | 1.26 | 0.4481 |
| unclipped versus full-clipped | 0.24 | 0.9678 |
| half-clipped versus full-clipped | −1.39 | 0.3852 |
| asymmetry of frequency (Hz) | | |
| unclipped versus half-clipped | −1.46 | 0.3527 |
| unclipped versus full-clipped | −0.33 | 0.9427 |
| half-clipped versus full-clipped | 1.47 | 0.3480 |
| asymmetry of angular velocity (rad s$^{-1}$) | | |
| unclipped versus half-clipped | 1.91 | 0.1920 |
| unclipped versus full-clipped | 1.39 | 0.3840 |
| half-clipped versus full-clipped | −0.74 | 0.7494 |

**Table 5.** Effect of clipping status as a fixed effect and body weight as a covariate on whole-body kinematics.

| whole-body kinematic measures | clipping status | | body weight | |
|---|---|---|---|---|
| | F-value | p-value | F-value | p-value |
| descent velocity (m s$^{-1}$) | 5.31 | 0.0395* | 2.72 | 0.1433 |
| descent angle ($\gamma$) | 5.05 | 0.0440* | 2.01 | 0.1992 |
| vertical acceleration (m s$^{-2}$) | 3.41 | 0.0924 | 0.52 | 0.4930 |
| horizontal acceleration (m s$^{-2}$) | 17.77 | 0.0018* | 3.86 | 0.0903 |

*Significance at $p < 0.05$.

half-clipped hens. The initial half of flight in the half-clipped hens had a vertical velocity of 1.44 ± 0.33 m s$^{-1}$ and the second half was 3.57 ± 0.36 m s$^{-1}$.

While vertical acceleration was not significantly affected by wing clipping (table 5), we observed a trend for increased vertical acceleration in the half-clipped hens compared to the full-clipped birds

**Table 6.** Comparison of whole-body kinematics between unclipped, half-clipped and full-clipped groups.

| whole-body kinematic measures | t-value | p-value |
| --- | --- | --- |
| descent velocity (m s$^{-1}$) | | |
| unclipped versus half-clipped | 1.87 | 0.2168 |
| unclipped versus full-clipped | −0.64 | 0.8031 |
| half-clipped versus full-clipped | −3.24 | 0.0337* |
| descent angle ($\gamma$) | | |
| unclipped versus half-clipped | 1.75 | 0.2526 |
| unclipped versus full-clipped | −0.71 | 0.7655 |
| half-clipped versus full-clipped | −3.16 | 0.0372* |
| vertical acceleration (m s$^{-2}$) | | |
| unclipped versus half-clipped | −1.55 | 0.3260 |
| unclipped versus full-clipped | 0.45 | 0.8940 |
| half-clipped versus full-clipped | 2.59 | 0.0817 |
| horizontal acceleration (m s$^{-2}$) | | |
| unclipped versus half-clipped | 5.21 | 0.0031* |
| unclipped versus full-clipped | 5.69 | 0.0018* |
| half-clipped versus full-clipped | 0.08 | 0.9968 |

*Significant difference at $p < 0.05$.

(table 6). The vertical acceleration of half-clipped hens was 4.59 m s$^{-2}$ (table 2), which translates to the ability to support only 46.79% of their body weight (vertical acceleration/gravitational force (9.81 m s$^{-2}$)). The unclipped and full-clipped hens had a vertical acceleration of 2.62 and 2.08 m s$^{-2}$, supporting 73.29 and 79.8% of their body weight, respectively. Horizontal acceleration was significantly affected by clipping treatment (table 5), whereby both the half- and full-clipped hens exhibited significantly lower values than unclipped hens (table 6). The clipped birds decelerated horizontally almost five times faster than the unclipped birds.

## 4. Discussion

We predicted that increasing wing loading by symmetrically clipping the wings would cause hens to alter wingbeat frequency, amplitude ($\theta$) and angular velocity to maintain flight trajectories. Our results indicate that laying hen wing kinematics do not change in response to significant wing area losses up to 50% (tables 3 and 4). Our data suggest that fully feathered laying hens are already at their performance limit for slow-speed, descending flight. None of the groups under study demonstrated an ability to support more than 80% of their body weight, with the half-clipped hens supporting less than 50% of their weight. This result reveals that the flight capacity of domesticated, egg-laying hens is poor. They may be generating maximum power output from their primary flight muscles (pectoralis and supracoracoideus) [43] such that they are unable to increase wingbeat frequency or amplitude to increase angular velocity and generate more power. Due to domestication, egg-laying chickens have a significantly higher wing loading than expected for their body weight [9]. Indeed, they are much larger, and exhibit decreased flight capabilities compared to their closest living ancestor, the red jungle fowl (*Gallus gallus*) [9,44,45]. Tobalske & Dial [29] report that flight performance declines, and the pectoralis muscle strain increases, with increasing body size in Phasianidae. Therefore, wing kinematic measurements that show little to no change following an increase in wing loading may indeed be a characteristic of Phasianidae birds with naturally high wing loadings and limited flight capabilities, including domestic laying hens.

One potential limitation of this study is the timing of the jump tests, as they were not conducted immediately after wing-feather clipping due to a larger ongoing experiment. The seven-week time gap between clipping and testing may have allowed the hens to acclimatize and adapt to their new wing loading and adjust their flapping patterns in additional ways that were not measured, such as the

angle of attack (the wing angle relative to the stroke plane). As this is the first study of its kind in laying hens, we chose to use kinematic measurements that are most representative of power output and widely used in wild bird literature. There were no changes in body weight within treatment groups between the two-week weigh-in intervals except for the full-clipped hens between initial wing clipping and the second week ($p = 0.0086$). However, no further significant differences in weight were exhibited for the full-clipped hens in the subsequent weeks, indicating that their weight stabilized over that time.

Although the hens did not modulate wing kinematics in response to clipping, we observed several differences in the whole-body kinematic measurements. Compared to their fully clipped counterparts, half-clipped hens' descent angle ($\gamma$) was significantly lower, allowing the birds to travel farther outward before landing. Similarly, the half-clipped hens' descent velocity was significantly lower than the full-clipped hens. This non-progressive result may be explained by individual variation among the birds, especially one potential outlier hen in the half-clipped group. She exhibited the lowest descent angle recorded of 31.3°, which represents almost a 14° deviation from the next lowest descent angle of 45.2°. Due to the small sample size of the present study, it is possible that this individual skewed the results. Consequently, a broader sampling may indicate no effect of wing clipping on descent angle.

Similarly, the half-clipped hens showed a trend towards a greater vertical acceleration relative to their unclipped or full-clipped counterparts, suggesting that half-clipped birds are not as efficient at supporting their body weight against gravitational force. It is unclear why the half-clipped group exhibited the highest vertical acceleration and, consequently, supported the smallest body-weight proportion.

The vertical component of descent velocity varied between 2.41 and 3.57 m s$^{-1}$ among treatment groups, and these are two to three times the landing speeds that pigeons (*Columba livia*), zebra finch (*Taeniopygia guttata*) and diamond doves (*Geopelia cuneata*) use just prior to touchdown on a perch [46–48]. These other species are smaller in mass than hens, so we highlight that hens landing at greater velocities have proportionally greater kinetic energy at touchdown that could contribute to injury [8,49].

All birds exhibited negative horizontal acceleration, indicating that they decelerated as they approached landing. Surprisingly, both half- and full-clipped hens decelerated at a rate almost five times faster than that of the unclipped hens. Birds with a larger wingspan and wing area should exhibit greater deceleration as their larger surface area produces more reverse thrust and drag [48]. This principle is exemplified in peregrine falcons that can meticulously control their deceleration while diving by extending their wings to increase the wingspan [50]. The hens' potential motivation to slow down and stop as they were descending may account for this unexpected observation. Most birds landed close to the wall opposite the jump tower in the enclosed test arena.

Studies in other bird species have reported that asymmetric wing clipping causes a change in wing kinematics, whereas symmetric wing clipping, as in our study, does not induce any changes [51–53]. Asymmetric wings can produce unequal torques from disproportionate flapping forces that may impact the birds' muscular and skeletal system, causing uneven strain [51,54]. The asymmetric distribution of lift forces can significantly impact a bird's manoeuvring capabilities [55], and zoological facilities often implement asymmetric clipping to limit escape capacity in open-air bird exhibits [56].

While the study protocol required hens to jump from a height of 1.5 m, hens in large aviary systems are often required to navigate spaces that are several metres high. Domestic chickens have also demonstrated to be less adept at navigating downwards versus upwards [57]. Therefore, it is possible that a more significant height challenge may have revealed differences in wing kinematics based on clipping status. Wing feather loss has also been shown to impair the detection of position and movements and inhibit the ability to regain balance through wing-flapping [58]. Therefore, it may still contribute to diminished locomotor and navigational abilities that eventually impact keel bone damage in domestic birds. Because keel bone damage is so widespread among laying hens [10], it may be that their near maximal power output while airborne, even when fully feathered, poses a risk to all laying hens when attempting to descend from higher elevations. Their immense landing velocities compared to other birds at landing, coupled with their large body sizes, increases their risk of landing or crashing with great amounts of kinetic energy. As a result, it may behove hen caretakers to ensure that aviaries and housing systems do not necessitate hens to make long downward journeys through the air to obtain resources.

# 5. Conclusion

Domestic laying hens did not exhibit differences in their wing kinematics, despite a dramatic increase in wing loading due to clipping of flight feathers. This suggests that fully feathered laying hens are near

maximal power output during flapping descent, thereby making changes to wing kinematics, for increased power output, impossible despite the significant reduction in wing area. To improve animal welfare, our result highlights the need to address fundamental questions regarding the link between feather loss, locomotor behaviour and navigation skills in domestic birds.

Ethics. This study was approved by the University of Guelph Animal Care Committee (Animal Utilization Protocol no. 3908) before being conducted.

Data accessibility. The data are provided in electronic supplementary material [59].

Authors' contributions. B.M.L. performed the experiment; carried out video and data analyses; drafted the manuscript; B.W.T. conceived and designed the experiment; participated in video and data analysis and helped draft the manuscript; N.B.S. performed the experiment and helped draft the manuscript; R.G. performed the experiment and helped draft the manuscript; D.R.P. conceived and designed the experiment and helped draft the manuscript; A.H.-M. conceived and designed the experiment; coordinated the study; participated in data analysis and helped draft the manuscript; all authors gave final approval for publication.

Competing interests. We have no competing interests.

Funding. This research was funded by Agriculture and Agri-Food Canada, AgriScience Program Cluster Activity no. 17-1 and by the Ontario Ministry of Agriculture, Food and Rural Affairs (OMAFRA) no. 27371.

Acknowledgements. We thank Arkell Research Station animal caretakers for taking excellent care of the birds.

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
