## [Peer Review File · Royal Society Open Science]

Review History

RSOS-210196.R0 (Original submission)

Review form: Reviewer 1

Is the manuscript scientifically sound in its present form?

Yes

Are the interpretations and conclusions justified by the results?

Yes

Is the language acceptable?

Yes

Do you have any ethical concerns with this paper?

No

Have you any concerns about statistical analyses in this paper?

No

Recommendation?

Accept with minor revision (please list in comments)

Comments to the Author(s)

The authors present data relevant to the role of high wing loading of domestic chickens on the risk of injury during descents from elevated roosts. They find that experimental manipulations leading to significant increases of wing loading via wing feather clipping changed very little about the full body or wing kinematics of hens descending from elevated perches.

While the data and conclusions may seem, at first, to be limited in relevance to the husbandry of domestic chickens, the authors successfully provide enough context to make the findings relevant to studies of allometry and wing design in wild birds. After all, the Origin of Species relied heavily on domesticated species, and I think there is room to include domestic chickens here.

The paper is well-prepared and very clear. I have few minor comments and suggestions for consideration. First, I have a single methodological concern, and another regarding results and interpretations.

Most of the wing kinematics are measured from the single camera with a frontal view. The camera was elevated substantially above the floor, presumably to provide a view orthogonal to the stroke plane of the descending birds. However, if the birds in different treatments significantly modified their stroke planes, or descended at very different angles, stroke amplitude measurements from that frontal camera may be affected by some degree of parallax or distortion. Can the side view be used to at least provide confidence that the stroke plane angle was consistent enough among treatments, and/or that the frontal camera was close enough to orthogonal that the stroke amplitude measurements are comparable?

In the results and discussion, the authors present "descent velocity", which I believe from the methods represents total speed. However it seems to me that the vertical component of landing speed, the speed right before touch down, is the most relevant to injury. The other measures may be helpful in understanding the biomechanical flight consequences of increased wing loading, but they don't necessarily provide insight to the birds' motivation. The authors effectively make the assumption that the birds are trying to minimize injury by minimizing landing velocity (especially the vertical component). That leads to the interpretation that the lack of changes of wing kinematics with wing clipping suggests that the muscles are already providing maximal power, and can't compensate. Alternatively, the results could reflect a lack of motivation to slow further than they are. That leads to two considerations. First, these domestic birds are heavier than their wild ancestors (and probably more-recent pre-industrial domestic ancestors), and free from the selection against injury. Might it be possible that their brains haven't fully caught up to their more-injury prone size? So if the vertical component of velocity at touch down is still "safe", they either don't need to, or don't realize they need to, slow down more.

For a free falling object from 150 cm we would expect it to hit with about 5.4 m/s. Many wild birds of various sizes land with 1.5-2.5 m/s. And the overall descent velocity is about 1.3-1.4 for all the groups. So it may be sufficiently slow, even with the clipping, that they are not overly "concerned" with reducing it further (though my own chickens sound like a watermelon hitting a wood floor when they drop from their roosts). Vertical velocity is even calculated in the matlab code, so that should be easy to include.

Specific comments:

page numbers refer to the page numbers at the top of the reviewer proof (intro starts on page 3)

P3 L35-48: It's never made explicit in this hypothesis, but I'm assuming that these fractures are thought to result from the birds dropping off of elevated roosts to the ground? I think that should be explicitly stated.

There, and in the next paragraph, the authors use "manoeuvring" without, I think, sufficient definition or consistency. Most readers unfamiliar with biomechanical analyses visualize manoeuvring as changes in direction, when much of this manuscript is dealing more with linear manoeuvring, i.e. linear accelerations or changes in speed.

As a result of those previous two comments, the introduction never makes explicit the hypothesized connections from high wing loading and reduced ability to resist gravity to higher collision speed with the ground to increased injuries.

P4L16-23: Phasianidae also, sometimes, use flight to fly up to elevated roosts even in the absence of predators. I think that point is particularly relevant for this context. My "domestic" guinea fowl roost 40 feet up in trees if I let them.

P4L24 "...impacts flapping performance...". Should this be "flapping-flight performance"? I think most measures here are really flight performance while flapping, rather than the performance of flapping itself.

P9L8-12: This is also consistent with my alternative interpretation of a lack of motivation. They've already had time to acclimate to the wings and "realized" that their landing speed was sufficiently slow.

P9L26-29: I'm confused as to the physics (or statistics) of this. How is it possible that the half-clipped birds had both the highest vertical acceleration (suggesting tending towards free-fall), and the most horizontal descent angles? Those seem to contradict each other.

Looking at the matlab code, I see that the velocities used to calculate accelerations are calculated with the square root of the square difference, effectively taking the absolute value. This is fine if the birds always move in the same direction. However, if they jumped up, and had an upward vertical velocity early, then a downward vertical velocity later, this calculation method would underestimate the vertical acceleration.

Review form: Reviewer 2

Is the manuscript scientifically sound in its present form?

Yes

Are the interpretations and conclusions justified by the results?

Yes

Is the language acceptable?

Yes

Do you have any ethical concerns with this paper?

No

Have you any concerns about statistical analyses in this paper?

No

Recommendation?

Accept with minor revision (please list in comments)

Comments to the Author(s)

Comments are provided in the attached document (see Appendix A).

Decision letter (RSOS-210196.R0)

Dear Dr Harlander-Matauschek

On behalf of the Editors, we are pleased to inform you that your Manuscript RSOS-210196 "Domestic egg-laying hens, *Gallus gallus domesticus*, do not modulate flight performance in response to wing condition" has been accepted for publication in Royal Society Open Science subject to minor revision in accordance with the referees' reports. Please find the referees' comments along with any feedback from the Editors below my signature.

Please submit your revised manuscript and required files (see below) no later than 7 days from today's (ie 07-May-2021) date. Note: the ScholarOne system will 'lock' if submission of the revision is attempted 7 or more days after the deadline. If you do not think you will be able to meet this deadline please contact the editorial office immediately.

on behalf of Kevin Padian (Subject Editor)

Subject Editor Comments to Author (Professor Kevin Padian):

Comments to the Author:

Thanks for your attention to the reviewers' comments. There are a few additional suggestions that appear helpful, and we ask you to address them in your final version. Best wishes.

Reviewer comments to Author:

Reviewer: 1

Comments to the Author(s)

The authors present data relevant to the role of high wing loading of domestic chickens on the risk of injury during descents from elevated roosts. They find that experimental manipulations leading to significant increases of wing loading via wing feather clipping changed very little about the full body or wing kinematics of hens descending from elevated perches.

While the data and conclusions may seem, at first, to be limited in relevance to the husbandry of domestic chickens, the authors successfully provide enough context to make the findings relevant to studies of allometry and wing design in wild birds. After all, the Origin of Species relied heavily on domesticated species, and I think there is room to include domestic chickens here.

The paper is well-prepared and very clear. I have few minor comments and suggestions for consideration. First, I have a single methodological concern, and another regarding results and interpretations.

Most of the wing kinematics are measured from the single camera with a frontal view. The camera was elevated substantially above the floor, presumably to provide a view orthogonal to the stroke plane of the descending birds. However, if the birds in different treatments significantly modified their stroke planes, or descended at very different angles, stroke amplitude measurements from that frontal camera may be affected by some degree of parallax or distortion. Can the side view be used to at least provide confidence that the stroke plane angle was consistent enough among treatments, and/or that the frontal camera was close enough to orthogonal that the stroke amplitude measurements are comparable?

In the results and discussion, the authors present "descent velocity", which I believe from the methods represents total speed. However it seems to me that the vertical component of landing speed, the speed right before touch down, is the most relevant to injury. The other measures may be helpful in understanding the biomechanical flight consequences of increased wing loading, but they don't necessarily provide insight to the birds' motivation. The authors effectively make the assumption that the birds are trying to minimize injury by minimizing landing velocity (especially the vertical component). That leads to the interpretation that the lack of changes of wing kinematics with wing clipping suggests that the muscles are already providing maximal power, and can't compensate. Alternatively, the results could reflect a lack of motivation to slow further than they are. That leads to two considerations. First, these domestic birds are heavier than their wild ancestors (and probably more-recent pre-industrial domestic ancestors), and free from the selection against injury. Might it be possible that their brains haven't fully caught up to their more-injury prone size? So if the vertical component of velocity at touch down is still "safe", they either don't need to, or don't realize they need to, slow down more.

For a free falling object from 150 cm we would expect it to hit with about 5.4 m/s. Many wild birds of various sizes land with 1.5-2.5 m/s. And the overall descent velocity is about 1.3-1.4 for all the groups. So it may be sufficiently slow, even with the clipping, that they are not overly "concerned" with reducing it further (though my own chickens sound like a watermelon hitting a

wood floor when they drop from their roosts). Vertical velocity is even calculated in the matlab code, so that should be easy to include.

Specific comments:

page numbers refer to the page numbers at the top of the reviewer proof (intro starts on page 3)

P3 L35-48: It's never made explicit in this hypothesis, but I'm assuming that these fractures are thought to result from the birds dropping off of elevated roosts to the ground? I think that should be explicitly stated.

There, and in the next paragraph, the authors use "manoeuvring" without, I think, sufficient definition or consistency. Most readers unfamiliar with biomechanical analyses visualize manoeuvring as changes in direction, when much of this manuscript is dealing more with linear manoeuvring, i.e. linear accelerations or changes in speed.

As a result of those previous two comments, the introduction never makes explicit the hypothesized connections from high wing loading and reduced ability to resist gravity to higher collision speed with the ground to increased injuries.

P4L16-23: Phasianidae also, sometimes, use flight to fly up to elevated roosts even in the absence of predators. I think that point is particularly relevant for this context. My "domestic" guinea fowl roost 40 feet up in trees if I let them.

P4L24 "...impacts flapping performance...". Should this be "flapping-flight performance"? I think most measures here are really flight performance while flapping, rather than the performance of flapping itself.

P9L8-12: This is also consistent with my alternative interpretation of a lack of motivation. They've already had time to acclimate to the wings and "realized" that their landing speed was sufficiently slow.

P9L26-29: I'm confused as to the physics (or statistics) of this. How is it possible that the half-clipped birds had both the highest vertical acceleration (suggesting tending towards free-fall), and the most horizontal descent angles? Those seem to contradict each other.

Looking at the matlab code, I see that the velocities used to calculate accelerations are calculated with the square root of the square difference, effectively taking the absolute value. This is fine if the birds always move in the same direction. However, if they jumped up, and had an upward vertical velocity early, then a downward vertical velocity later, this calculation method would underestimate the vertical acceleration.

Reviewer: 2

Comments to the Author(s)

Comments are provided in the attached document

===PREPARING YOUR MANUSCRIPT===

===PREPARING YOUR REVISION IN SCHOLARONE===

- Any electronic supplementary material (ESM).
- If you are requesting a discretionary waiver for the article processing charge, the waiver form must be included at this step.
- If you are providing image files for potential cover images, please upload these at this step, and inform the editorial office you have done so. You must hold the copyright to any image provided.
- A copy of your point-by-point response to referees and Editors. This will expedite the preparation of your proof.

- Ensure that your data access statement meets the requirements at <https://royalsociety.org/journals/authors/author-guidelines/#data>. You should ensure that you cite the dataset in your reference list. If you have deposited data etc in the Dryad repository, please only include the 'For publication' link at this stage. You should remove the 'For review' link.
- If you are requesting an article processing charge waiver, you must select the relevant waiver option (if requesting a discretionary waiver, the form should have been uploaded at Step 3 'File upload' above).
- If you have uploaded ESM files, please ensure you follow the guidance at <https://royalsociety.org/journals/authors/author-guidelines/#supplementary-material> to include a suitable title and informative caption. An example of appropriate titling and captioning may be found at https://figshare.com/articles/Table_S2_from_Is_there_a_trade-off_between_peak_performance_and_performance_breadth_across_temperatures_for_aerobic_scope_in_teleost_fishes_/3843624.

Author's Response to Decision Letter for (RSOS-210196.R0)

See Appendix B.

Decision letter (RSOS-210196.R1)

Dear Dr Harlander-Matauschek,

I am pleased to inform you that your manuscript entitled "Domestic egg-laying hens, *Gallus gallus domesticus*, do not modulate flapping flight performance in response to wing condition" is now accepted for publication in Royal Society Open Science.

If you have not already done so, please remember to make any data sets or code libraries 'live' prior to publication, and update any links as needed when you receive a proof to check - for

instance, from a private 'for review' URL to a publicly accessible 'for publication' URL. It is good practice to also add data sets, code and other digital materials to your reference list.

You can expect to receive a proof of your article in the near future. Please contact the editorial office (openscience@royalsociety.org) and the production office (openscience_proofs@royalsociety.org) to let us know if you are likely to be away from e-mail contact – if you are going to be away, please nominate a co-author (if available) to manage the proofing process, and ensure they are copied into your email to the journal. Due to rapid publication and an extremely tight schedule, if comments are not received, your paper may experience a delay in publication.

on behalf of Prof Kevin Padian (Subject Editor)
openscience@royalsociety.org

Appendix A

This paper attempts to understand the impact of feather loss on domestic egg-laying hens' flight capabilities and how is their wing kinematics affected due to the increased wing loading as a result of feather loss. The authors conducted descending flight (jump) experiments with 18 adult hens and used two high-speed video cameras to record the flight and to extract the wing kinematics. They clipped the primary and secondary feathers to replicate the impact of feather loss. According to their results, the hens did not exhibit any differences in their wing kinematics, despite a dramatic increase in wing loading due to clipping of flight feathers. The authors suggest that laying hens operate at the maximal power output and have limited capability to improve their flight for increased power output. Overall, the article is well written and abstract. Though the article does not provide distinguishing results and the limitations are addressed by the authors in the discussion. This article is interesting as it attempted to analyze the effect of wing kinematics due to changes in wing loading in the flight of Phasianidae birds, which are under-explored. I have few minor concerns below that must be addressed before publication:

Comments:

1) In the summary/abstract, the authors state the following “We expected birds to compensate for increased wing loading by increasing wingbeat frequency, amplitude, and angular velocity. Otherwise, we expected to observe an increase in descent velocity and angle and a decrease in horizontal and vertical acceleration.” These statements by the authors are logical and true, but I feel no adequate discussion is provided to arrive at this research anticipation either in the introduction or discussion. How or why these factors are expected to change as indicators of power output or flight capability? This should be discussed from basic principles and supported using literature.

2) The authors stated that the jump tests were conducted seven weeks after wing clipping. Were the birds weighed regularly? How much percentage of the birds' weight changed over that period and what was the implication on wing loading over the same period? Authors further state “.....have allowed the hens to acclimatize and adapt to their new wing load and adjust their flapping patterns in additional ways that were not measured, such as the angle of attack”. In such a case, it has a significant impact on the authors' claim that the hens did not exhibit differences in their wing kinematics, despite a dramatic increase in wing loading. Therefore, other unmeasured kinematic parameters might have varied. Can the authors explain how the role of other factors may not have a better implication on the increased power output?

- 3) Line 46: “Thus, a reduction in wing load”. Higher wing loading affects maneuverability. The start of the sentence should be changed either to ‘a reduction in wing area’ or ‘an increase in wing loading’ due to the loss of the feathers. Also, I suggest providing more relevant citation here than the 15.

- 4) Line 30: Wingbeat frequency as an indicator of power output is supported by citation number 17, but not the wingbeat amplitude and angular velocity. Additional support for the other two factors should be included.

- 5) Bird training: how were the birds trained? were the birds jumped on their own volition immediately once placed into the start box and platform assembly. Few details on these should be included.

- 6) I suggest following wing loading consistently throughout the text than alternating with wing load

- 7) Line 46: Figure 3 is referred to in the text but no information is provided. The purpose of figure 3 should be discussed.

- 8) Line 49: it will be useful to add a schematic representation to show how the wingbeat amplitude is calculated.

- 9) The authors mentioned quite in-detail about keel-bone injuries due to increased wing loading in the introduction and research question. But it wasn’t addressed later in their discussion about what their results implicate on the keel-bone injuries.

Appendix B

Reviewer comments are listed in quotation marks and author responses are shown beneath each comment in bold type.

Reviewer 1

1. “The paper is well-prepared and very clear. I have few minor comments and suggestions for consideration. First, I have a single methodological concern, and another regarding results and interpretations.”

Most of the wing kinematics are measured from the single camera with a frontal view. The camera was elevated substantially above the floor, presumably to provide a view orthogonal to the stroke plane of the descending birds. However, if the birds in different treatments significantly modified their stroke planes, or descended at very different angles, stroke amplitude measurements from that frontal camera may be affected by some degree of parallax or distortion. Can the side view be used to at least provide confidence that the stroke plane angle was consistent enough among treatments, and/or that the frontal camera was close enough to orthogonal that the stroke amplitude measurements are comparable?

We agree that deviation from orthogonal placement of the frontal-view camera would contribute unwanted error in our measures of wingbeat amplitude. Using our observed variation in descent angle, the use of sines to estimate true length with a deviation angle due to observed descent angle, we estimate this unwanted error to be $\leq 4\%$. We have added this new description to the text in the methods

2. “In the results and discussion, the authors present "descent velocity", which I believe from the methods represents total speed. However it seems to me that the vertical component of landing speed, the speed right before touch down, is the most relevant to injury. The other measures may be helpful in understanding the biomechanical flight consequences of increased wing loading, but they don't necessarily provide insight to the birds' motivation. The authors effectively make the assumption that the birds are trying to minimize injury by minimizing landing velocity (especially the vertical component). That leads to the interpretation that the lack of changes of wing kinematics with wing clipping suggests that the muscles are already providing maximal power, and can't compensate. Alternatively, the results could reflect a lack of motivation to slow further than they are. That leads to two considerations. First, these domestic birds are heavier than their wild ancestors (and probably more-recent pre-industrial domestic ancestors), and free from the selection against injury. Might it be possible that their brains haven't fully caught up to their more-injury prone size? So if the vertical component of velocity at touch down is still "safe", they either don't need to, or don't realize they need to, slow down more.
For a free falling object from 150 cm we would expect it to hit with about 5.4 m/s. Many wild birds of various sizes land with 1.5-2.5 m/s. And the overall descent velocity is about 1.3-1.4 for all the groups. So it may be sufficiently slow, even with the clipping, that they are not overly "concerned" with reducing it further (though my own chickens sound like a

watermelon hitting a wood floor when they drop from their roosts). Vertical velocity is even calculated in the matlab code, so that should be easy to include.”

This is a useful and interesting line of reasoning, and we now include vertical velocity during the latter half of flight, thus prior to touchdown, as text in the results. We observed values that varied from 2.46 - 3.57 m/s. These are 2-3x greater than landing velocities, just prior to touchdown, in several species (pigeons, diamond doves and zebra finch) for which there are detailed kinematic measurements during landing. This pattern, coupled with the larger size of hens, underscores the risks domestic layer hens experience due to high kinetic energy at touchdown. Because the velocities for hens are greater than for species that are more adept at flight, we interpret that the hen velocities are not well within a safety factor.

3. “P3 L35-48: It's never made explicit in this hypothesis, but I'm assuming that these fractures are thought to result from the birds dropping off of elevated roosts to the ground? I think that should be explicitly stated.

There, and in the next paragraph, the authors use "manoeuvring" without, I think, sufficient definition or consistency. Most readers unfamiliar with biomechanical analyses visualize manoeuvring as changes in direction, when much of this manuscript is dealing more with linear manoeuvring, i.e. linear accelerations or changes in speed.

As a result of those previous two comments, the introduction never makes explicit the hypothesized connections from high wing loading and reduced ability to resist gravity to higher collision speed with the ground to increased injuries.”

We have added some potential causes of keel bone damage for background, as well as expanded on the implications of our hypothesis and prediction in the introduction.

4. “P4L16-23: Phasianidae also, sometimes, use flight to fly up to elevated roosts even in the absence of predators. I think that point is particularly relevant for this context. My "domestic" guinea fowl roost 40 feet up in trees if I let them.”

We have added “and to roost” at the end of the sentence

5. “P4L24 "...impacts flapping performance...". Should this be "flapping-flight performance"? I think most measures here are really flight performance while flapping, rather than the performance of flapping itself.”

We have ensured that the phrasing is consistent throughout the paper

6. “P9L8-12: This is also consistent with my alternative interpretation of a lack of motivation. They've already had time to acclimate to the wings and "realized" that their landing speed was sufficiently slow.”

This has been addressed with the interpretation of the new descent velocity results, explained below.

7. “P9L26-29: I'm confused as to the physics (or statistics) of this. How is it possible that the half-clipped birds had both the highest vertical acceleration (suggesting tending towards free-fall), and the most horizontal descent angles? Those seem to contradict each other.”

We thank the reviewer for bringing this to our attention. Upon re-evaluating our MATLAB code, we found an error in which we did not include a time component for overall descent velocity. We have carefully redone our calculations to ensure accuracy. This changed our results for descent velocity only, and the manuscript has been updated accordingly. This edit does not change the overall results of the study, and in fact strengthens the argument that laying hens have limited flight capacity. To address the reviewers comment, the half-clipped hens had an initial vertical velocity lower than the un- and full-clipped birds, then increased their landing vertical velocity to greater than the other two groups. The resultant shape of this flapping descent is the hens initially traveled farther outward with a low velocity and angle, then landed with a very high velocity. We have included explanations of the vertical components of descent velocity into the results and discussion text.

8. “Looking at the matlab code, I see that the velocities used to calculate accelerations are calculated with the square root of the square difference, effectively taking the absolute value. This is fine if the birds always move in the same direction. However, if they jumped up, and had an upward vertical velocity early, then a downward vertical velocity later, this calculation method would underestimate the vertical acceleration.”

We have added a statement to the Materials and Methods addressing this. No birds in the videos used had an initial jump before descending.

Reviewer 2

1. “ In the summary/abstract, the authors state the following “We expected birds to compensate for increased wing loading by increasing wingbeat frequency, amplitude, and angular velocity. Otherwise, we expected to observe an increase in descent velocity and angle and a decrease in horizontal and vertical acceleration.” These statements by the authors are logical and true, but I feel no adequate discussion is provided to arrive at this research anticipation either in the introduction or discussion. How or why these factors are expected to change as indicators of power output or flight capability? This should be discussed from basic principles and supported using literature.”

We have added additional statements to the introduction to make this connection clearer.

2. “The authors stated that the jump tests were conducted seven weeks after wing clipping. Were the birds weighed regularly? How much percentage of the birds’ weight changed over that period and what was the implication on wing loading over the same period? Authors further state “.....have allowed the hens to acclimatize and adapt to their new wing load and adjust their flapping patterns in additional ways that were not measured, such as the angle of attack”. In such a case, it has a significant impact on the authors’ claim that the hens did not exhibit differences in their wing kinematics, despite a dramatic increase in wing loading. Therefore, other unmeasured kinematic parameters might have varied. Can the authors explain how the role of other factors may not have a better implication on the increased power output?”

We have included a statement about body weight. Overall, we can say that weight loss was not significant between feather clipping and testing, and thus their wing loading remained constant between that time as well. Their wing loading at clipping was the same as at testing. We have also added justification for why we chose to use our kinematic measures and not others.

3. “ Line 46: “Thus, a reduction in wing load”. Higher wing loading affects maneuverability. The start of the sentence should be changed either to ‘a reduction in wing area’ or ‘an increase in wing loading’ due to the loss of the feathers. Also, I suggest providing more relevant citation here than the 15.”

We thank the reviewer for pointing out this error, it has been corrected.

4. “Line 30: Wingbeat frequency as an indicator of power output is supported by citation number 17, but not the wingbeat amplitude and angular velocity. Additional support for the other two factors should be included.”

We have included additional citations.

5. “Bird training: how were the birds trained? were the birds jumped on their own volition immediately once placed into the start box and platform assembly. Few details on these should be included.”

We have added information on this topic to the Materials and Methods.

6. “I suggest following wing loading consistently throughout the text than alternating with wing load”

We have checked for consistent use and changed when applicable

7. “Line 46: Figure 3 is referred to in the text but no information is provided. The purpose of figure 3 should be discussed.”
8. “Line 49: it will be useful to add a schematic representation to show how the wingbeat amplitude is calculated.”

The two above comments were addressed by adding markings to Figure 3 to indicate the top of the upstroke and bottom of the downstroke, thus expanding on the purpose of Figure 3 by incorporating a schematic representation of wingbeat amplitude.

9. “The authors mentioned quite in-detail about keel-bone injuries due to increased wing loading in the introduction and research question. But it wasn’t addressed later in their discussion about what their results implicate on the keel-bone injuries.”

The discussion has been expanded to include a more robust discussion of the results and their implications on keel bone damage.